# Evaluation of Cr(VI) Reduction Using Indigenous Bacterial Consortium Isolated from a Municipal Wastewater Sludge: Batch and Kinetic Studies

**Buyisile Kholisa \*, Mpumelelo Matsena**  **and Evans M. N. Chirwa** 

Water Utilisation and Environmental Engineering Division, Department of Chemical Engineering, University of Pretoria, Pretoria 0002, South Africa; mpumelelo.matsena@gmail.com (M.M.); evans.chirwa@up.ac.za (E.M.N.C.)
* Correspondence: buyisile.kholisa@tuks.co.za

**Abstract:** Hexavalent Chromium (Cr(VI)) has long been known to be highly mobile and toxic when compared with the other stable oxidation state, Cr(III). Cr(VI)-soluble environmental pollutants have been detected in soils and water bodies receiving industrial and agricultural waste. The reduction of Cr(VI) by microbial organisms is considered to be an environmentally compatible, less expensive and sustainable remediation alternative when compared to conventional treatment methods, such as chemical neutralization and chemical precipitation of Cr. This study aims to isolate and identify the composition of the microbial consortium culture isolated from waste activated sludge and digested sludge from a local wastewater treatment plant receiving high loads of Cr(VI) from an abandoned chrome foundry in Brits (North Waste Province, South Africa). Furthermore, the Cr(VI) reduction capability and efficiency by the isolated bacteria were investigated under a range of operational conditions, i.e., pH, temperature and Cr(VI) loading. The culture showed great efficiency in reduction capability, with 100% removal in less than 4 h at a nominal loading concentration of 50 mg Cr(VI)/L. The culture showed resilience by achieving total removal at concentrations as high as 400 mg Cr(VI)/L. The consortia exhibited considerable Cr(VI) removal efficiency in the pH range from 2 to 11, with 100% removal being achieved at a pH value of 7 at a 37 ± 1 °C incubation temperature. The time course reduction data fitted well on both first and second-order exponential rate equation yielding first-order rate constants in the range 0.615 to 0.011 h$^{-1}$ and second order rate constants 0.0532 to $5 \times 10^{-5}$ L·mg$^{-1}$·h$^{-1}$ for Cr(VI) concentration of 50–400 mg/L. This study demonstrated the bacterial consortium from municipal wastewater sludge has a high tolerance and reduction ability over a wide range of experimental conditions. Thus, show promise that bacteria could be used for hexavalent chromium remediate in contaminated sites.

**Keywords:** sludge cultures; culture characterization; Cr(VI) reduction kinetics; soil remediation; phylogenetic analysis; toxic metalloids

## 1. Introduction

Chromium (Cr) and its compounds have been extensively used in many industrial processes, such as metal finishing, metal electroplating, steelworks manufacturing, wood preservation, leather tanning, textile dyeing, and synthesis of pigments [1,2]. As a result of the wide anthropogenic use of Cr, large quantities of Cr containing wastes have been produced, and the lack of effective disposal methods of Cr effluents has led to the contamination of surface and groundwater environments, soils and aquatic sediments [3]. Another major concern is that these high Cr(VI) effluents end up in municipal sewer lines and build up in the sludge because only a small quantity is discharged with the wastewater final effluent [4]. The application of municipal sludge in agricultural soils poses health risk threats. Cr is mainly present in the environment in two oxidation states: trivalent (Cr(III)) and hexavalent (Cr(VI)) species [5]. Cr(VI) has been recognized as more hazardous because

of its higher solubility and mobility, its rapid permeability, and strong oxidizing ability, which exert harmful effects on biological systems [6,7]. Consequently, USEPA has set the allowable limit for Cr(VI) in domestic water at 0.05 mg/L, and 0.01 mg/L for aquatic life [8]. In comparison to Cr(VI), Cr(III) is an important microelement for sustaining human metabolism and homeostasis. It is less toxic and readily forms highly insoluble hydroxide/oxides in the environment at a pH value higher than 5.5 [2,9,10]. Thus, reduction of Cr(VI) to the relatively nonhazardous Cr(III) is an effective strategy to mitigate the risks for human health and the environment.

There are various conventional technologies available for minimizing the environmental impact of Cr(VI), including chemical reduction, ion exchange, electrochemical treatment, membrane separation, etc. [11–13]. However, most of these technologies are often ineffective and very expensive, especially for low concentration of metals [14]. Additionally, the use of chemical reagents produces an enormous amount of hazardous sludge that requires further treatment [15]. Therefore, it is essential to develop an innovative, cost-effective, and environmental friendly alternative process to remediate Cr(VI) contamination.

The bioreduction of toxic Cr(VI) to less toxic Cr(III) using microbial organisms is considered as a valuable, promising, and cost-effective approach for Cr(VI) remediation. The first case of microbial reduction of Cr(VI) was reported in the late 1970s by Romanenko & Koren'Kov (1977), where isolated Pseudomonas strain was tested. Since then, numerous scholars have isolated new Cr(VI)-reducing microbial strains under different conditions, such as Bacillus [16–18], Pseudomonas [19–21], Microbacterium [22], Desulfovibrio [23], Enterobacter [24,25], Halomonas [26], and Escherichia [27]. Various environments, such as industrial landfills, waste disposal sites, coal mines, as well as tannery effluents and contaminated sediments from rivers, have been identified as main target areas to isolate these potential strains for in situ bioremediation.

Liu et al. [28] studied a Bacillus sp. isolated from a Cr landfill site for the reduction of Cr(VI) at an initial Cr(VI) concentration of 80 mg/L, and observed a maximum of 81.5% reduction of Cr(VI) in 72 h. Banerjee et al. [29] isolated Bacillus cereus strain from an open-cast coal mine that completely reduced a 200 mg/L Cr(VI) concentration within 16 h under heterotrophic conditions. Wani et al. [20] isolated pseudomonas species from Cr(VI)-contaminated alloy manufacturing effluent and evaluated its Cr(VI) reduction performance. They observed a maximum Cr(VI) reduction of 86% at 100 mg/L under neutral pH conditions, and a 120 h incubation time. Li et al. [30] studied the treatment of high-concentration chromium-containing wastewater using sulfate-reducing bacteria acclimated with ethanol under various conditions. Their results showed that the strain was capable of reducing Cr(VI) concentration up to 500 mg/L under the optimum pH value of 7, a temperature of 35 °C, and an incubation time of 24 h, and the volume amount of chromium-containing wastewater to bacteria was 5:1. The distinctions in the reduction capacities of bacterial strains are avowed to be directly dependent on the physicochemical parameters and the heavy metal concentration in various environmental conditions. Consequently, exploration for native bacterial systems for in situ bioremediation of that specific polluted location is always valuable.

This work aims to isolate and investigate the Cr(VI) reduction by consortium bacterium from a sludge coming from a municipal wastewater treatment of Brits, South Africa, that receives high periodic loads of hexavalent chromium from a chrome foundry nearby. Our previous study, Molokwane et al. [3], showed that sludge bacteria from the Brits plant had high Cr(VI) reduction capacity. The current study assesses how the Cr(VI) initial concentration, the initial pH solution, and co-existing heavy metals affect the removal of Cr(VI) by consortia bacteria from the wastewater treatment plant. Furthermore, bacterium Cr(VI) reduction kinetics were also studied. This is an effort to expand the development of bioremediation technique for Cr(VI) treatment of polluted sites in South Africa, since South Africa holds the largest chrome ore reserves in the world and it is one of the largest producers of ferrochrome [31].

## 2. Results and Discussion

### 2.1. Bacteria Screening for Cr(VI) Reduction

The three sludge samples collected from different locations within the wastewater treatment plant were used a source of indigenous microbial consortia, and were screened and examined for their ability to reduce Cr(VI). The Cr(VI)-reducing ability was tested for each microbial consortia at a Cr(VI) concentration ranging from 100 mg/L to 500 mg/L. These experiments were carried out in LB broth supplemented with Cr(VI). All three microbial consortia showed good Cr(VI)-reducing capability, as shown in Table 1. It can be seen that complete Cr(VI) reduction of 100 mg/L Cr(VI) initial concentration was achieved by all the microbial consortia. However, as the initial Cr(VI) concentration was increased, Cr(VI) reduction decreased accordingly, and all the microbial consortia only managed a reduction of less than 7% at the highest initial Cr(VI) concentration of 500 mg/L. The loss of Cr(VI) reduction capacity by the microbial consortia was due to Cr(VI) inhibition. Sludge C microbial consortia exhibited more Cr(VI) reducing power than sludge A and B microbial consortia at higher concentrations. This was due to better acclimation and longer exposure to Cr(VI), and sludge C it is a combination of sludge A and B. Therefore, microbial consortia from sludge C was than preferred for further studies.

**Table 1.** % Cr(VI) reduction by microbial consortia from primary sludge (sludge A), activated sludge (sludge B), and dry sludge (sludge C) under different initial Cr(VI) concentration incubated for 24 h.

| Cr(VI) Concentration (mg/L) | Percentage Reduction | | |
|:---:|:---:|:---:|:---:|
| | Sludge A | Sludge B | Sludge C |
| 100 | 100 | 100 | 100 |
| 200 | 60.7 | 65.4 | 57.2 |
| 300 | 25.9 | 29.9 | 20.1 |
| 400 | 4.7 | 4.5 | 11.4 |
| 500 | 0.29 | 5.0 | 6.2 |

### 2.2. Abiotic Controls

To examine the abiotic reduction of Cr(VI) by heat-killed and azide-inhibited cells, experiments were conducted using a Cr(VI) concentration of 100 mg/L. In these experiments, the extent of Cr(VI) removal was evaluated from the changes in Cr(VI) concentration in the aqueous phase. Figure 1 illustrates the extent of Cr(VI) reduction in a batch system containing cells-free, heat-killed and azide-inhibited cells. After 24 h of operation time, it was observed that only 4, 10, and 17% of Cr(VI) was removed from thet cell-free, heat-killed, and azide-inhibited cells, respectively. The low Cr(VI) reduction with heat-killed cells was due to the inactivation of the cells by heat. The 10% Cr(VI) removed by heat-killed cells was due to the existence of live cells that survived the heat destruction. Cells inhibited with azide, also showed low Cr(VI) reduction capability due to cells inactivation under oxygen-stressed environment. These results indicate that Cr(VI) reduction by live cells was not due abiotic factors.

### 2.3. Effect of Cr(VI) Concentration

The effect of initial Cr(VI) concentration on Cr(VI) reduction was studied over a range of 50–400 mg/L at a constant pH and temperature of 7.2 and 37 °C, respectively, under aerobic conditions. As shown in Figure 2, the bacteria consortia could completely reduce Cr(VI) concentration of 50 mg/L within 5 h of incubation. As Cr(VI) initial concentration increased it took longer for the bacteria to completely reduce the Cr(VI), as 100, 150, and 200 mg/L were reduced in 18, 72, and 96 h, respectively. However, above a 200 mg/L Cr(VI) initial concentration, complete Cr(VI) reduction was not observed, as it can be seen that 300 and 400 mg/L concentrations were reduced by 92% and 68%, respectively, after 120 h when the experiment was terminated. The slow reduction capabilities at high concentrations can be ascribed to the Cr(VI) reduction bacteria reaching the Cr(VI)

toxicity level. Molokwane et al. [3], and Wang and Shen [32], showed that the loss of Cr(VI) reduction capacities by bacteria was due to the loss of cell viability at high Cr(VI) concentration. These results show that Cr(VI) toxicity does significantly affect the Cr(VI) reduction by microorganisms.

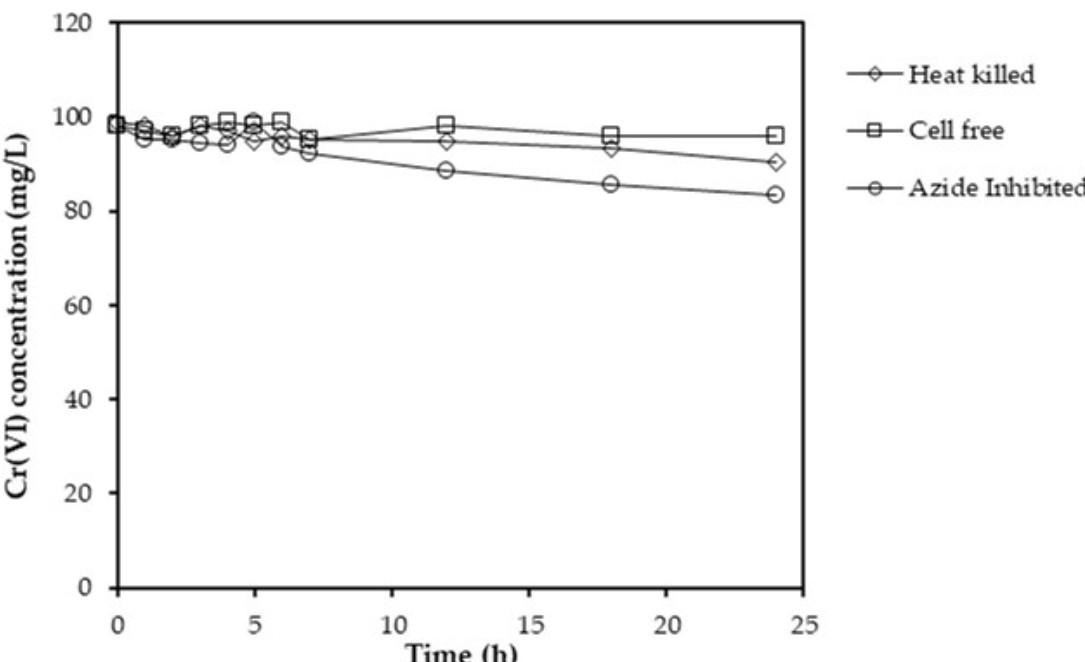

**Figure 1.** Evaluation of abiotic reduction of Cr(VI) by heat-killed and azide inhibited reducing cells under aerobic conditions.

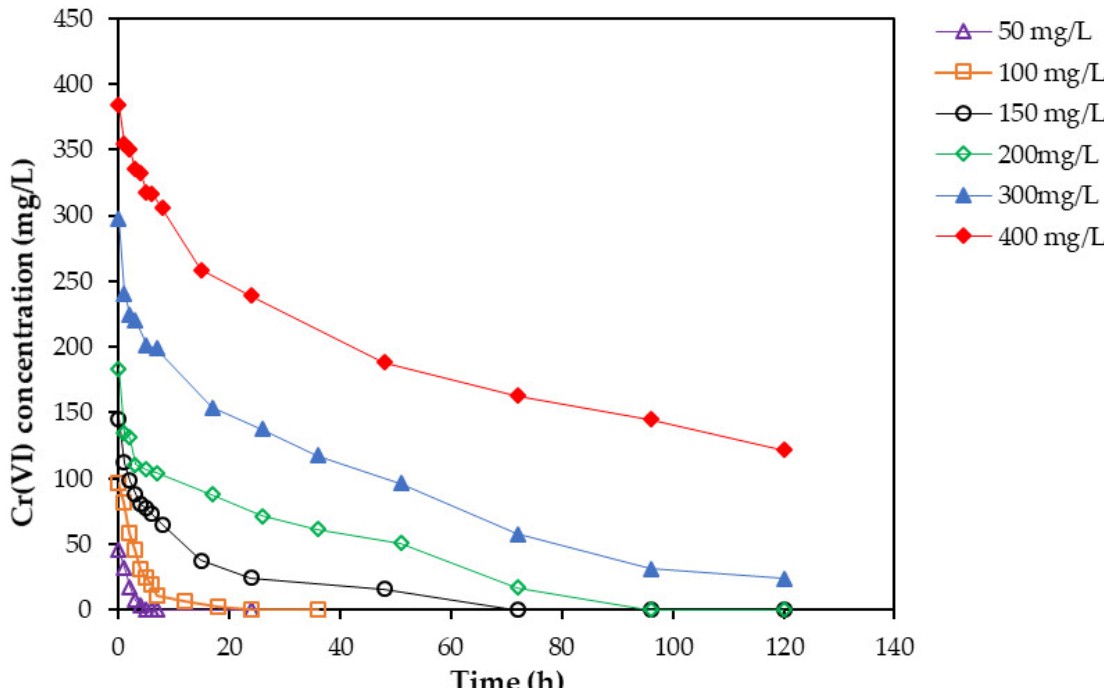

**Figure 2.** Effect of initial Cr(VI) concentration on Cr(VI) reduction using indigenous consortia bacteria.

### 2.4. Bacteria Performance at Different Cr(VI) Concentrations

The specific Cr(VI) reduction rates determined after 5 h, and the overall specific reduction at varying initial Cr(VI) concentrations, are given in Figure 3a,b. However, the overall specific Cr(VI) reduction rate increased with the increasing initial Cr(VI) concentration until a peak was reached at 300 mg/L and a further increase of the Cr(VI) concentration to

400 mg/L resulted in a decrease of the overall specific rate, suggesting a possible Cr(VI) inhibition. These results show that the Cr(VI) reduction process is catalyzed by microbial consortia saturation kinetics. Similar kinetic patterns have been reported for the Bacillus strain [33]; the Hypocrea tawa strain [34]; and Shewanella oneidensis MR-1 [35]. However, in these studies, the overall specific Cr(VI) reduction rate was determined in terms of the protein concentration and the particulate organic carbon. Zakaria et al. [36], and Jeyasingh and Philip [37], also indicate that, even though they did not observe complete Cr(VI) reduction, the initial specific reduction rate increased with Cr(VI) concentration.

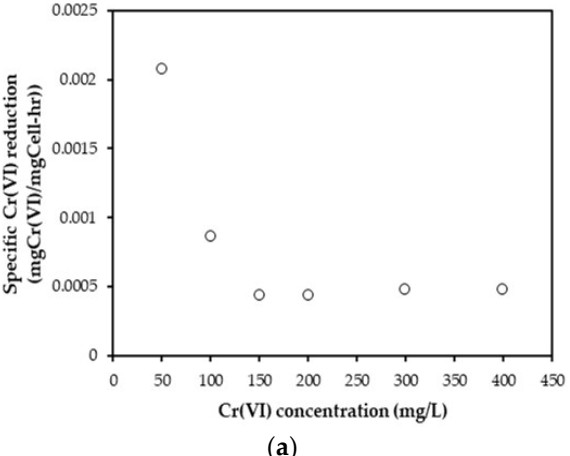
(a)

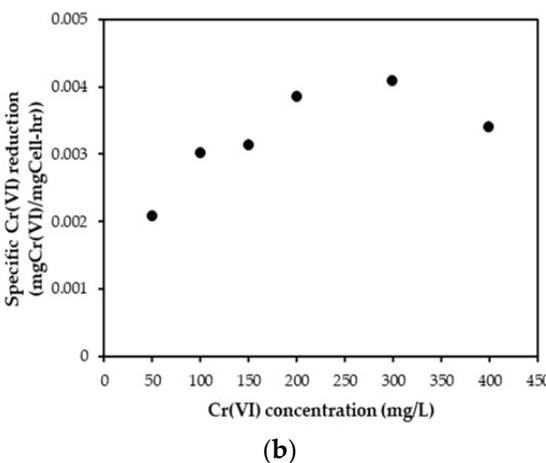
(b)

**Figure 3.** Specific Cr(VI) reduction at varying Cr(VI) concentration (**a**) after 5 h (**b**) after the duration of the experiment.

The specific Cr(VI) reduction rate is defined as a measure of Cr(VI) reduction per unit mass of biomass per hour. It can be seen that the specific Cr(VI) reduction rate after 5 h decreased with the increasing initial Cr(VI) concentration, reaching a minimum of 0.00043 mg Cr(VI)/mg biomass h at 150 mg/L, and remained constant at a Cr(VI) concentration higher than 150 mg/L. However, the overall specific Cr(VI) reduction rate increased with the increasing initial Cr(VI) concentration until a peak was reached at 300 mg/L and a further increase of the Cr(VI) concentration to 400 mg/L resulted in a decrease of the overall specific rate, suggesting a possible Cr(VI) inhibition. These results show that the Cr(VI) reduction process is catalyzed by microbial consortia saturation kinetics. Similar kinetic patterns have been reported for the Bacillus strain [33]; the Hypocrea tawa strain [34]; and Shewanella oneidensis MR-1 [35]. However, in these studies, the overall specific Cr(VI) reduction rate was determined in terms of the protein concentration and the particulate organic carbon. Zakaria et al. [36], and Jeyasingh and Philip [37], also indicate that, even though they did not observe complete Cr(VI) reduction, the initial specific reduction rate increased with Cr(VI) concentration.

### 2.5. Effect of pH on Cr(VI) Reduction

The pH of the solution is an important parameter, it can affect the activity of the bacteria, the degree of enzyme ionization, and the accessibility of heavy metal ions [18,38,39]. Hence, the efficiency of the microbial removal of heavy metals is affected. The Cr(VI) residual concentration by indigenous consortia bacterium results at different pH levels are presented in Figure 4. Cr(VI) reduction by consortia bacterium was studied over a range of 2–11 pH levels in an MSM medium amended with 50 mg/L Cr(VI), and incubated at 37 °C under aerobic conditions (shown in Figure S1). As the initial pH increased, Cr(VI) residuals showed a decrease from the pH of 2 to 7, followed by an increasing residuals from pH 7 to 10. The consortia showed an enhanced removal efficiency at neutral pH, and near complete was observed within 5 h. Acidic and alkaline conditions severely inhibited Cr(VI) reduction by the indigenous bacteria consortia from wastewater sludge. These results highlight that Cr(VI) removal by indigenous consortia bacterium was higher in neutral to acidic

conditions, as compared alkaline conditions. The widespread pH adaptability and efficient Cr(VI) removal ability under neutral–acidic conditions suggest that indigenous consortia bacterium could play a significant role in the bioremediation of acidic Cr polluted sites.

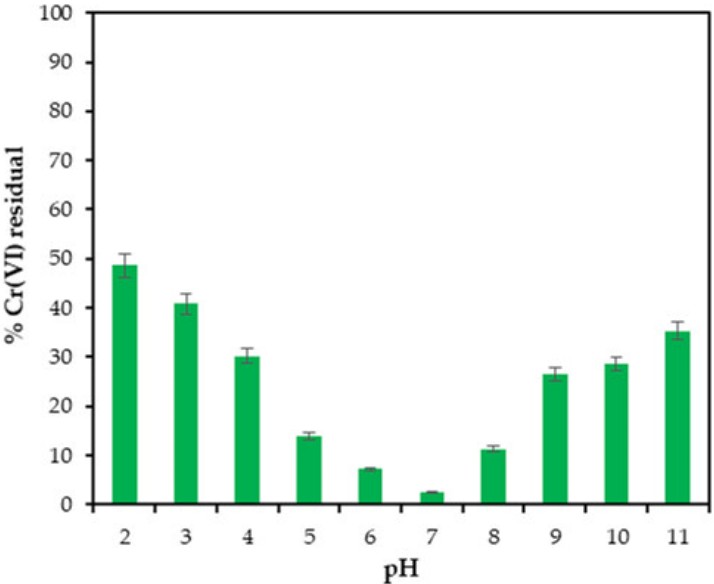

**Figure 4.** Effect of pH medium used for Cr(VI) reduction by indigenous consortia bacterium from wastewater sludge.

### 2.6. Effects of Coexisting Heavy Metals on Cr(VI) Reduction

Contaminated groundwater, soils, and industrial and municipal wastewater usually contain other heavy metals, and the presence of these coexisting heavy metal ions may have an effect on the Cr(VI) reduction by microorganisms. Thus, the effect of coexisting heavy metals on Cr(VI) reduction by mixed bacteria consortium was studied in this work (Figure 5). The influence of heavy metals on Cr(VI) reduction was studied using five metals ($Ni^{2+}$, $Cu^{2+}$, $Zn^{2+}$, $Mn^{2+}$ and $Pb^{2+}$) at 5 and 50 mg/L concentrations, and the Cr(VI) concentration was fixed at 50 mg/L. The microbial consortia completely reduced Cr(VI) within 5 h in the absence of other heavy metals. The presence of 5 mg/L of $Ni^{2+}$, $Mn^{2+}$, and $Pb^{2+}$ had no effect on Cr(VI) reduction. However, an enhanced reduction rate was observed with $Cu^{2+}$ and $Zn^{2+}$, as the Cr(VI) reduction was complete within 2 h and 3 h, respectively. However, enhanced Cr(VI) reduction was achieved in the presence of $Cu^{2+}$ and $Zn^{2+}$, with completely Cr(VI) reduction being observed within the first hour of incubation, while with $Zn^{2+}$, was achieved in 3 h at 50 mg/L metal concentration. Several studies have reported the induced Cr(VI) reduction by microbial organisms in the presence of $Cu^{2+}$, including Bacillus sp. CRB-B1 strain, Bacillus strain TCL, and Acinetobacter haemolyticus [18,29,36]. The cause of the stimulating effect of $Cu^{2+}$ and other metals on Cr(VI) reduction by microbial organisms is not yet clear. According Tan et al. [18], and Huang et al. [40], the increase in Cr(VI) reduction caused by $Cu^{2+}$ is due to the fact that it is one of the essential components of some antioxidants, such as superoxide dismutase and catalase. Moreover, it acts as an electron transporter for oxidative respiratory system [41]. At a higher heavy metal concentration of 50 mg/L, a significant inhibition of Cr(VI) reduction was observed in the presences of $Ni^{2+}$ and $Pb^{2+}$, with 81 and 43% Cr(VI) reduction being achieved, respectively, whereas the presence of $Mn^{2+}$ had no effect. Similar results were reported by Bhattacharya and Gupta [42], the presence of $Ni^{2+}$ and $Pb^{2+}$ significantly inhibited chromate reduction by Acinetobacter sp. B9. The inhibition of Cr(VI) removal by some heavy metals at elevated concentrations maybe due to the suppression of microbial activity by metal toxicity, and the destruction of protein structures by heavy metals [14].

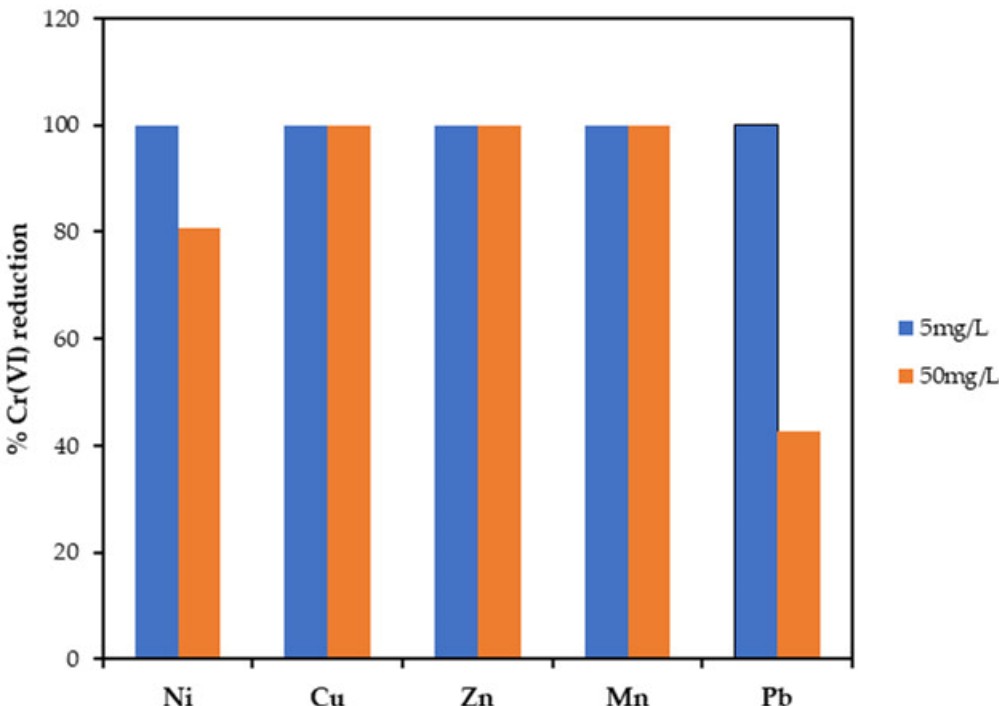

**Figure 5.** The effect of 5 and 50 mg/L of various heavy metals on reduction of 50 mg/L Cr(VI) by wastewater sludge bacteria consortia.

### 2.7. Kinetics of Cr(VI) Reduction by Bacteria Consortia

To quantitatively determine the interaction between the reduction rate and time for Cr(VI) removal by the bacterium consortia coming from a wastewater treatment plant, the kinetics for Cr(VI) reduction were conducted. The kinetics of the Cr(VI) bioreduction at varying initial Cr(VI) concentrations was studied by first- and second-order exponential decay [11,17,39–42].

$k_1$ was determined as the slope from plotting $\ln([Cr(VI)]/[Cr(VI)_0])$ versus time. The estimated $k_1$ values, and their coefficient of determination $R^2$ values resulting from linear regression, are given in Table 2. As shown in Figure 6a and Table 2, the fitting $R^2$ of the initial concentration of 50, 100, 200, 300, and 400 mg/L were 0.96, 0.93, 0.82, 0.94, and 0.86 respectively, indicating that the first-order exponential decay accurately described the reduction process of Cr(VI) over time. The $k_1$ was found to decrease (from 0.615 $h^{-1}$ to 0.011 $h^{-1}$) with increasing Cr(VI) concentration from 50 mg/L to 400 mg/L. Similar results to this study have been reported by Das et al. [43] and Tan et al. [18]. However, their Cr(VI) reduction rate values were less by two order of magnitude. This was due to the differences in experimental conditions, such as the Cr(VI) concentrations, the reduction medium, and the bacterial strains. Cr(VI)-reducing bacterial strains have different reduction capabilities due to different mechanisms used tolerate/reduce Cr(VI).

**Table 2.** First and second order kinetic of Cr(VI) reduction by bacterial consortia and their correlation coefficient.

| Cr(VI) Concentration | Pseudo-First Order | | Pseudo-Second Order | |
|---|---|---|---|---|
| | $k_1$ | $R^2$ | $k_2$ | $R^2$ |
| 50 | 0.615 | 0.96 | 0.0532 | 0.74 |
| 100 | 0.225 | 0.93 | 0.0152 | 0.85 |
| 150 | 0.056 | 0.69 | 0.0012 | 0.99 |
| 200 | 0.032 | 0.82 | 0.0005 | 0.79 |
| 300 | 0.023 | 0.94 | 0.0003 | 0.91 |
| 400 | 0.011 | 0.86 | 0.00005 | 0.97 |

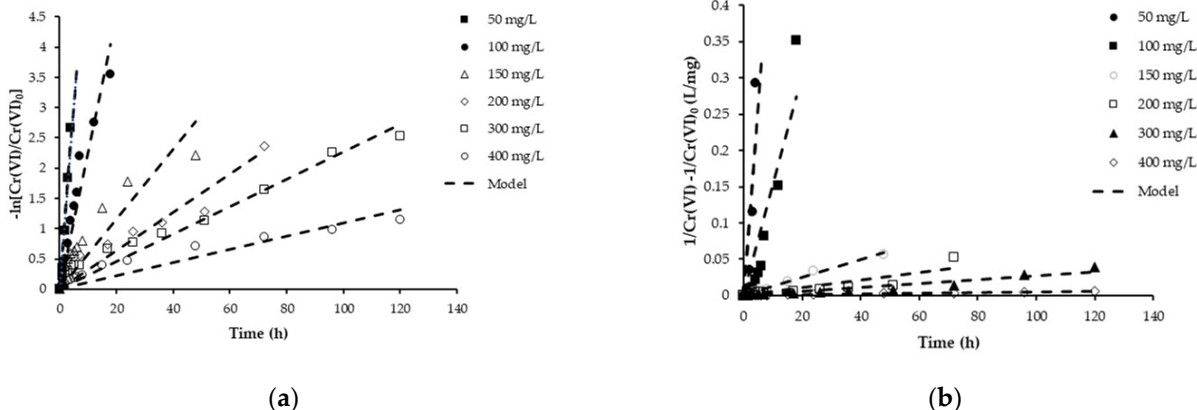

**Figure 6.** The kinetics of Cr(VI) reduction by bacterial consortia at different Cr(VI) initial concentration (**a**) pseudo-first-order kinetics (**b**) second-order kinetics.

The second-order rate constant $k_2$ was determined as the slope from plotting $1/[Cr(VI)-1/[Cr(VI)_0]$ versus time. The second-order rate constants $k_2$, and their coefficient of determination $R^2$ values resulting from linear regression are given in Table 2. The fitting $R^2$ of different initial concentration 100, 150, 200, 300, and 400 mg/L were 0.85, 0.99, 0.79, 0.91, and 0.97, respectively, indicating that the second-order exponential decay accurately described the reduction process of Cr(VI) over time as illustrated in Figure 6b. However, 50 mg/L proved to be less efficient than the first-order exponential model. The $k_2$ values followed a similar trend as $k_1$, and were found to be decreasing from $0.0532-5 \times 10^{-5}$ $L \cdot mg^{-1} \cdot h^{-1}$ with increasing Cr(VI) concentration of 50 mg/L to 400 mg/L. Even though the overall Cr(VI) reduction rate declined with the increasing Cr(VI) initial concentration, the total amount of Cr(VI) removed at higher initial concentrations was greater after 120 h (Table S1).

### 2.8. Microbial Characterization

Sludge C bacteria consortium was chosen for characterization due to its high performance. The bacterial isolates were identified based on 16S rDNA gene sequencing analyses, and were carried out by the Microbiology Department, at the University of Pretoria, to identify bacterial communities present after the sludge had been exposed to 100 mg/L of Cr(VI). BLASTN analyses of the bacterial isolates, X1, X2, X3, X4, X5, X6, and X7, are presented in Table 3, and show four predominant species under aerobic conditions. The sequence for X1 was 99% similar to that of Bacillus cereus 213 16S and the Bacillus thuringiensis strains. The X2 and X3 isolates produced similar results and showed a close association with B. cereus ATCC 10987, Bacillus sp. ZZ2 16s, and B. thuringiensis str. Al Hakam having a 99% identity, while X4, X5, and X6 were in close association with B. mycoides strain BGSC 6A13 16S, and the B. thuringiensis serovar finitimus strain BGSC 4B2 16S strains. The sequence for X7 was 99% similar to that of Microbacterium sp. S15-M4 and Microbacterium foliorum. A phylogenetic tree was constructed for the species from purified cultures grown under aerobic conditions based on a basic BLAST search of rRNA sequences in the NCBI database (Figure 7).

Bacteria that are able to reduce and tolerate Cr(VI) have been reported by many researchers and, primarily, these microorganisms are found in chromium contaminated sites. Soni et al. [44] isolated four bacteria strains from soil irrigated with tannery wastewater: Bacillus sp., Microbacterium sp., Bacillus thuringiensis, and Bacillus subtilis, all were able to reduce Cr(VI) at varying concentrations. Upadhyay et al. [45] reported that Bacillus sp. MNU16 isolated from a coal contaminated mine was able to tolerate and reduce Cr(VI). Banerjee et al. [29] also showed that Bacillus cereus MBGIPS 9 from a coal mine lake has a high tolerance for Cr(VI) toxicity and reduction capacity.

**Table 3.** Sludge Cr(VI)-Reducing Bacteria strain characterisation using 16S rRNA [3].

| Blast Results | Pure Isolates | | | | | | | ID Index |
|---|---|---|---|---|---|---|---|---|
| | X1 | X2 | X3 | X4 | X5 | X6 | X7 | |
| B. cereus ATCC 10987 | - | √ | √ | - | - | - | - | 0.99 |
| B. thuringiensis serovar finitimus strain BGSC 4B2 16S | - | - | - | √ | √ | √ | - | 0.99 |
| B. thuringiensis str. Al Hakam | - | √ | √ | - | - | - | - | 0.99 |
| Bacillus cereus strain 213 16S | √ | - | - | - | - | - | - | 0.99 |
| Bacillus mycoides strain BGSC 6A13 16S | - | - | - | √ | √ | √ | - | 0.99 |
| Bacillus sp. ZZ2 16S | - | √ | √ | - | - | - | - | 0.99 |
| Bacillus thuringiensis 16S | √ | - | - | - | - | - | - | 0.99 |
| Microbacterium foliorum | - | - | - | - | - | - | √ | 0.99 |
| Microbacterium sp. S15-M4 | - | - | - | - | - | - | √ | 0.99 |

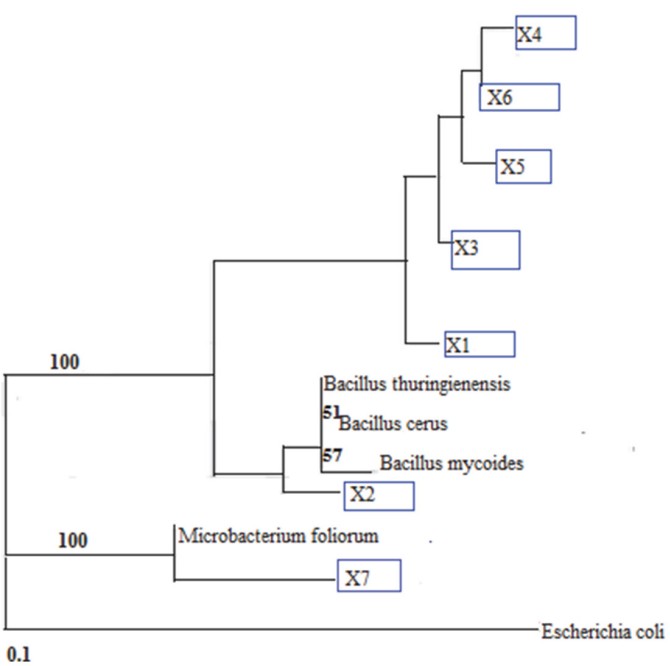

**Figure 7.** Phylogenetic tree constructed by neighbor-joining algorithm based on the partial 16S rRNA gene sequences and 1000 bootstrap replicates, showing the microbial diversity of Cr(VI) reducing consortium from Sludge C under aerobic conditions.

## 3. Materials and Methods

### 3.1. Chemical Reagents

All chemicals, reagents, and microbial media were analytically pure and procured from Sigma-Aldrich Pty Ltd. (Industrial Park, Jet Park, Johannesburg, South Africa) and additional purification was not required. A 1000 mg/L Cr(VI) concentration stock solution was prepared by dissolving $K_2CrO_4$ in deionized water and filter-sterilized. The Cr(VI) stock solution was diluted to the desired concentrations and used as a sources of Cr(VI). Diphenylcarbazide (Merck, South Africa) solution was prepared for Cr(VI) analysis by dissolving 0.5 g of 1,5-diphenylcarbazide in 100 mL of HPCL grade acetone and was stored in a brown bottle covered with a foil. 0.85%wt NaCl solution was prepared in distilled water. The mineral salt medium (MSM) consisted (g/L) of 2.12 $K_2HPO_4$, 2.12 $KH_2PO_4$, 2 NaCl, 1 $MgSO_4·7H_2O$, 0.1 $CaCl_2$, 4 $KNO_3$ and 5 D-glucose as a carbon source in distilled water [37].

Luria Bertani (LB) broth and agar were prepared according to manufacturer's instruction.

The pH was adjusted to 7 ± 0.2 by using HCl or NaOH. All mediums in this study were autoclaved at 121 °C for 15 min cooled to room temperature before use except for LB agar which was cooled to 40 °C.

### 3.2. *Bacteriul Culture*

#### 3.2.1. Source of Bacterial Culture

The natural bacteria consortia were obtained from sludge collected at the Brits Wastewater Treatment Works (North West Province, South Africa). An abandoned sodium dichromate processing facility was reported to discharge high levels of Cr(VI) periodically into the sewage treatment works. The chrome processing facility was commissioned as early as 1996. Three sludge samples were collected at different locations within the wastewater treatment plant, and were named as sludge A, B, and C. Sludge A is primary sludge, Sludge B is settled Activated Sludge collected from secondary clarifiers and Sludge C: is dewatered sludge from Sand Drying Beds–the dried product of the combined Primary Sludge. The samples were stored in sterile containers at 4 °C.

#### 3.2.2. Culture Screening and Cr(VI) Tolerance

The consortia bacterial cultures were screened for Cr(VI) reduction on the basis of their reduction performance under a free-oxygen environment. An amount of 1 g of sludge sample was added to a 250 mL conical flask containing 100 mL of LB broth and incubated for 24 h at 37 °C by agitation at 120 rpm using a Labcon SPL-MP 15 Lateral Shaker (Labcon Laboratory Services, Johannesburg, South Africa). The conical flask was closed with cotton to allow oxygen flow while preventing contaminants from entering the flask. After 24 h of incubation, 1 mL of this was transferred into a fresh 100 mL LB broth supplemented with 100 mg/L of Cr(VI). The fresh LB broth was incubated for 24 h under the same conditions. This method was replicated by gradually increasing the Cr(VI) concentration in the growth medium up to 500 mg/L.

#### 3.2.3. Microbial Characterization

Phylogenetic characterization and species identification of isolated bacteria cells were conducted on single colonies of bacteria grown aerobically from sludge sample. LB agar was used for colony development. Agar plates were inoculated with 1 mL samples and the colonies were subcultured using differential techniques (exhibited colours and morphologies), and incubated at 37 °C for 24 h. Bacteria colonies were first classified based on their morphologies, and then the 16S rRNA gene sequencing was performed. The genomic DNA of pure bacterial colonies was extracted by following the instructions of the manufacturer using a DNeasy tissue kit (QIAGEN Ltd., West Sussex, UK). The 16S rRNA genes of bacteria strains were carried out by reverse transcriptase-polymerase chain reaction (RT-PCR) amplification using pA and pH1 primers in the Microbiology Department at the University of Pretoria. The obtained sequence analysis data was compared to known bacteria in the GenBank using the online BLAST tool program in the National Centre for Biotechnology Information (NCBI).

### 3.3. *Batch Studies*

#### 3.3.1. Abiotic Experiments

To determine the abiotic reduction of Cr(VI), experiments were conducted under different conditions. Overnight grown cells were killed by heat then in the first flask, second flask was inhibited with sodium azide before inoculation. The third flask was inoculated without cells. Overnight cells were separated from the LB broth by centrifuging at 6500 rpm for 10 min, and then rinsing twice with a sterile saline solution (0.85% NaCl). The supernatant was decanted, and the cell pellet was used for the reduction experiments.

#### 3.3.2. Aerobic Reduction Experiments

The reduction of Cr(VI) by the freshly grown cells of the bacterial consortia was determined in the MSM. The experiments were conducted in a 250mL Erlenmeyer flask containing 100 mL MSM supplemented with 50–400 mg/L Cr(VI) concentration. The cells were harvested after a 24 h incubation and washed thrice by centrifugation with a 0.85% NaCl sterile solution, and finally resuspended in the MSM. Flasks were inoculated

with cells concentrated to a 5:1 ratio before adding Cr(VI), and the flasks were covered with cotton to allow oxygen while preventing microorganisms from entering. The flasks were incubated at 37 °C and under constant shaking at 120 rpm. All experiments were conducted in duplicate. The 1 mL samples were taken at time intervals determined by the observed rate of Cr(VI) removal. The samples were centrifuged at 6500 rpm for 10 min in a Hermle 2323 centrifuge (Hermle Laboratories, Wehigen, Germany) to remove suspended cells before analysis.

### 3.3.3. Effect of pH on Cr(VI) Reduction

The effect of different pH on Cr(VI) reduction was studied, with pH ranging from 2 to 11. The experiments were conducted at a 50 mg/L initial Cr(VI) concentration. The pH was adjusted by using HCl or NaOH.

### 3.3.4. Effect of Coexisting Heavy Metals on Cr(VI) Reduction

Polluted environment often contains other heavy metals and their existence may have an effect on the Cr(VI) reduction by microorganisms. The effect of heavy metals on Cr(VI) reduction was studied using five metals ($Ni^{2+}$, $Cu^{2+}$, $Zn^{2+}$, $Mn^{2+}$ and $Pb^{2+}$) at 5 and 50 mg/L concentrations, and Cr(VI) concentration was fixed at 50 mg/L.

### *3.4. Kinetic Parameter Estimation for Cr(VI) Reduction by Bacteria Consortia*

The kinetics of Cr(VI) reduction by sludge bacteria consortia were evaluated using first and second order rate laws.

### 3.4.1. First-Order Kinetics

The first-order kinetics assumes that when the Cr(VI) concentration decreases over time, the reaction rate also decreases linearly with Cr(VI) concentration. The rate of reaction (r) can be expressed using Equation (1):

$$r = -d[Cr(VI)]/dt = k_1[Cr(VI)] \tag{1}$$

Equation (1) is solved by integrating both side between the limits $[Cr(VI)_0]$ at t = 0 and [Cr(VI)] at any time t gives the following expression by Equation (2):

$$\ln([Cr(VI)]/Cr(VI)_0) = k_1 t \tag{2}$$

where, $k_1$ is the first order rate constant ($h^{-1}$), Cr(VI) is Cr(VI) concentration (mg/L) and $Cr(VI)_0$ is the initial concentration (mg/L).

### 3.4.2. Second-Order Kinetics

The second-order kinetics assumes that when the Cr(VI) concentration changes over time, the rate of reaction also changes proportional to the square of the Cr(VI) concentration. The rate of reaction can be expressed by Equation (3):

$$r = -d[Cr(VI)]/dt = k_2[Cr(VI)]^2 \tag{3}$$

Integrating Equation (3) between the limits $[Cr(VI)] = [Cr(VI)_0]$ at time t = 0 and [Cr(VI)] = [Cr(VI)] at any time t, yields the following Equation (4).

$$1/[Cr(VI)] = 1/[Cr(VI)_0] + k_2 t \tag{4}$$

where $k_2$ is the second-order rate constant ($L \cdot mg^{-1} \cdot h^{-1}$).

### *3.5. Analytical Methods*

1 mL samples were withdrawn periodically, then centrifuge at 6500 rpm for 10 min. An amount of 0.2 mL of the sample was added in a 10 mL volumetric flask, followed by 1 mL of 1 N $H_2SO_4$ and the addition of distilled water up to the 10 mL mark. An amount

of 0.2 mL of DPC solution was finally added to produce a purple colour. Cr(VI) was measured by a UV-Vis spectrophotometer (WPA, Light Wave II, Labotech, Johannesburg, South Africa) operated at a wavelength of 540 nm.

### 3.6. Biomass Analysis

To quantify initial biomass mass, a 5 mL sample was collected at the beginning of the experiment batches. The sample was centrifuged at 6500 rpm for 10 min. The pellet was resuspended in 1 mL distilled water and, afterwards it was passed through a Whatman filter paper (No. 1) with a pore size of 11 um. The filter paper containing biomass was oven dried and cooled until a constant weight was obtained at 75–80 °C. Biomass mass was determined as the difference between wet filter paper and the dried filter paper [46].

### 4. Conclusions

In this study, aerobic microbial mixed culture isolated from municipal wastewater treatment sludge was capable of reducing Cr(VI) at concentrations up to 400 mg/L. It was shown that Cr(VI) reduction was an enzyme-mediated process instead of an adsorption mediated one. A complete Cr(VI) reduction of 50 mg/L concentration was observed in 6 h under aerobic and neutral pH conditions. The Cr(VI) reduction rate decreases with increasing initial Cr(VI) concentration due to Cr(VI) toxicity on bacterial cells. It was observed that Cr(VI) reduction increases with increasing initial pH of the solution until the optimal pH of 7, and a further increase in the pH results in decreased Cr(VI) removal. Coexisting heavy metals did not have an effect in Cr(VI) reduction at both low and high heavy metal concentrations, with the exception of $Cu^{2+}$ and $Zn^{2+}$ which enhanced Cr(VI) reduction, while $Pb^{2+}$ and $Ni^{2+}$ showed an inhibitory effect at high concentrations. The fitting of the time course data to a first-order rate resulted in a rate constant in the range of $0.615 \ h^{-1}$ to $0.011 \ h^{-1}$, which decreased with an increasing Cr(VI) concentration from 50 to 400 mg/L. Similarly, the second-order rate constant was in the range of $0.0532–5 \times 10^{-5} \ L^{-1} \cdot mg^{-1} \cdot h$, and decreased with increasing initial concentration. The reduction ability of the mixed bacterial consortium to treat Cr(VI) may be explored further for practical application and developing a sustainable bioremediation process for Cr(VI)-contaminated areas. This is an effort to expand the development of bioremediation technique for the Cr(VI) treatment of polluted sites in South Africa.

**Supplementary Materials:** The following are available online at https://www.mdpi.com/article/10.3390/catal11091100/s1, Figure S1: Time course Cr(VI) reduction by bacterial consortia at different pH. Table S1: Total Cr and Cr(III) concentration measurements during Cr(VI) reduction using indigenous consortia bacteria.

**Author Contributions:** Conceptualization, B.K. and E.M.N.C.; methodology, B.K.; formal analysis, B.K. and M.M.; writing—original draft preparation, B.K.; data curation, B.K. and M.M.; writing— review and editing, E.M.N.C. and M.M.; investigation, B.K. and E.M.N.C.; funding acquisition, E.M.N.C.; resources and supervision, E.M.N.C. All authors have read and agreed to the published version of the manuscript.

**Funding:** This research was funded by the research was funded through the National Research Foundation (NRF) Grant No. IFR200206501999, CSUR210111581519, EQP210111581520 awarded to Prof Evans MN Chirwa of the University of Pretoria. Buyisile Kholisa would like to acknowledge the National Research Foundation (NRF) for granting him the Doctoral Scholarship, Grant No. MND190613447392.

**Acknowledgments:** The authors would like to thank Elmarie Otto and Alette Devega for their administrative support.

**Conflicts of Interest:** The authors declare no conflict of interest.

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
