# Peer review of "Evaluation of Cr(VI) Reduction Using Indigenous Bacterial Consortium Isolated from a Municipal Wastewater Sludge: Batch and Kinetic Studies"

_catalysts, doi:10.3390/catal11091100_

Round 1

Reviewer 1 Report

There should use dots instead of commas in the values of the figures - Table 2 and 3. 

Author Response

Thank you for pointing this out. The reviewer is correct, and we have changed the commas to dots in Tables 1, 2, 3 and Figures 3 and 6.

Reviewer 2 Report

The paper is aimed at reduction of Cr(VI) by microbial organisms. This study demonstrated the bacterial consortium from municipal wastewater sludge has a high tolerance and reduction ability over a wide range of experimental conditions. Thus, show promise that bacteria could be used for Cr(VI) remediate in contaminated area. he article is interesting, the introduction is written very well, introduces the reader to the research topic. The research was conducted in a thoughtful manner, the discussion of the obtained research results is of a good scientific level. The article can be published after taking into account the following remarks:

  • references in the article should be written in a uniform way, i.e. in some places they are written in superscript
  • should there be dots or commas in  Tables: 1, 2, 3 Figures: 3, 6 - it seems that there should be dots
  • suplementary materials are not attached - why?

Author Response

Reviewer 2

The paper is aimed at reduction of Cr(VI) by microbial organisms. This study demonstrated the bacterial consortium from municipal wastewater sludge has a high tolerance and reduction ability over a wide range of experimental conditions. Thus, show promise that bacteria could be used for Cr(VI) remediate in contaminated area. he article is interesting, the introduction is written very well, introduces the reader to the research topic. The research was conducted in a thoughtful manner, the discussion of the obtained research results is of a good scientific level. The article can be published after taking into account the following remark.

Author response: We would like to thank the reviewer for the kind words. We are excited about the prospect of publishing this article to contribute to the science and applications of Cr(VI) remediation in the environment.

  1. References in the article should be written in a uniform way, i.e. in some places they are written in superscript.

Author response: The reviewer is correct, and we have changed the intext references written in superscript in line 67.

  1. Should there be dots or commas in Tables: 1, 2, 3 Figures: 3, 6 - it seems that there should be dots.

Author response: Thank you for outlining this. The reviewer is correct, and we have changed the commas to dots in Tables 1, 2, 3 and Figures 3 and 6.

  1. supplementary materials are not attached - why?

Author response: Supplementary materials were added “Figure S1: Time course Cr(VI) reduction by bacterial consortia at different pH. Table S1: Total Cr and Cr(III) concentration measurements during Cr(VI) reduction using indigenous consortia bacteria”.

Reviewer 3 Report

The article “Evaluation of Cr(VI) reduction using indigenous bacterium consortia isolated from a Municipal Wastewater Sludge: Batch and kinetic studies” reports the results of a laboratory investigation aimed at assessing the capability of bacteria from municipal sludge of removing Cr(VI). Various Cr(VI) concentrations, pH, presence of other heavy metals are studied, together with kinetics analysis and microbial characterization. The article reports some novel content and can be relevant for Catalysts journal; however, there are several aspects to be improved before final publication. I consequently recommend a major revision that should cover the following aspects:

  1. Introduction: the issues related to heavy metals presence in wastewater (and sludge) should be briefly summarized, e.g. considering the strict limits for agricultural sludge reutilization (see 10.31025/2611-4135/2020.13993).
  2. Results and discussion, lines 96-98: this is valid only for high Cr(VI) concentrations, as it appears from Table 1 that sludge samples A and B are performing better until 300 mg/L of Cr(VI) concentration.
  3. Line 108: “illustrates” should be used instead of “illustrations”.
  4. Line 125: “completely” should substitute “complete”.
  5. Lines 139-143: the sentences are not grammatically correct. Please fix them.
  6. Line 147: the verb “is” is missing.
  7. Lines 175-177: this is true also for municipal wastewater.
  8. Lines 211-218 and 226-232 should be moved to Materials and methods section.
  9. Line 239: “at” should be removed.
  10. Table 2-3: the decimal separator should be “.”, not “,”.
  11. The discussion is very limited, especially considering the kinetics analysis and the microbial characterization. The authors should add some more comparison with existing literature also in these sections.
  12. What are the future perspectives of this study? The authors should briefly depict the path forward, especially in the Conclusions section.
  13. Section 3.1.2: is it possible to report some quantitative data about Cr(VI) concentration in the wastewater and in the sludge samples A, B, C?
  14. Line 359: what was filter pore diameter?
  15. Lines 380-395: these items are not part of the supplementary materials, as they are already present in the main text.
  16. The authors should better highlight the novelty of the present study compared to their previous work published in Water science and technology in 2008 (10.2166/wst.2008.669).
  17. English language needs to be improved as there are several errors throughout the manuscript. Native speaker revision would be beneficial.

Author Response

Reviewer 3

  1. Introduction: the issues related to heavy metals presence in wastewater (and sludge) should be briefly summarized, e.g. considering the strict limits for agricultural sludge reutilization (see 10.31025/2611-4135/2020.13993).

Author response: We have added the suggested content to the manuscript in lines 45-48. The revised text reads “Another major concern is that these high Cr(VI) effluents end up in municipal sewer lines and build up in the sludge because only a small quantity is discharged with the wastewater final effluent [4]. The application of municipal sludge in agricultural soils poses health risk threats.”  

  1. Results and discussion, lines 96-98: this is valid only for high Cr(VI) concentrations, as it appears from Table 1 that sludge samples A and B are performing better until 300 mg/L of Cr(VI) concentration.

Author response: We have amended lines 120-122 in the revised manuscript.  The amended text reads as follows “Sludge C microbial consortia exhibited more Cr(VI) reducing power than sludge A and B microbial consortia at higher concentrations.”

  1. Line 108: “illustrates” should be used instead of “illustrations”.

Author response: We have amended the text “illustrations” in line 133 to “illustrates”.

  1. Line 125: “completely” should substitute “complete”.

Author response: We have revised the text “complete” in line 147 to “completely”.

  1. Lines 139-143: the sentences are not grammatically correct. Please fix them.

Author response: We have revised the text in lines 164-167. The modified text reads “The specific Cr(VI) reduction rates determined after 5 h and the overall specific reduction at varying initial Cr(VI) concentration are given in Figure 3(a) and (b). The specific Cr(VI) reduction rate, which is defined as a measure of Cr(VI) reduction per unit mass of biomass per hour. It can be seen that the specific Cr(VI) reduction rate after 5 h decreased with increasing initial Cr(VI) concentration reaching a minimum of 0.00043 mg Cr(VI)/mg biomass h at 150 mg/L, and remained constant at Cr(VI) concentration higher than 150 mg/L.”

  1. Line 147: the verb “is” is missing.

Author response: We have inserted the verb “is” in line 174.

  1. Lines 175-177: this is true also for municipal wastewater.

Author response: We have inserted the text “municipal wastewater” in line 202

  1. Lines 211-218 and 226-232 should be moved to the Materials and methods section.

Author response: We have moved subsections 2.7.1 and 2.7.2 to a new section 3.4 Kinetic parameter estimation for Cr(VI) reduction by bacteria under materials and methods in lines 406-428.

  1. Line 239: “at” should be removed.

Author response: We have removed the text “at” in line 260.

  1. Table 2-3: the decimal separator should be “.”, not “,”.

Author response: Thank you for outlining this. The reviewer is correct, and we have changed the commas to dots in Tables 1, 2, 3 and Figures 3 and 6.

  1. The discussion is very limited, especially considering the kinetics analysis and the microbial characterization. The authors should add some more comparison with existing literature also in these sections.

Author response: We have taken the recommendation by the reviewer. The discussion was added in the kinetic analysis in lines 247-251 and microbial characterization in lines 293-300.

  1. What are the future perspectives of this study? The authors should briefly depict the path forward, especially in the Conclusions section.

Author response: We have added lines 462-463 in the conclusion section. The text reads “This is an effort to expand the development of bioremediation technique for Cr(VI) treatment of polluted sites in South Africa.”

  1. Section 3.1.2: is it possible to report some quantitative data about Cr(VI) concentration in the wastewater and in the sludge samples A, B, C?

Author response: In this study Cr(VI) measurements in the sludge was not done. However, in our previous study Molokwane et al. 2008, reported that Cr(VI) concentration in the influent and mixed liquor (biological reactor) from the treatment plant was 2.45 and 2.63 mg/L, respectively, and the Cr(VI) content in sludge C was 25.44 g/m3.

  1. Line 359: what was filter pore diameter?

Author response: We have added the text “with a pore size of 11 um” in line 440.

  1. Lines 380-395: these items are not part of the supplementary materials, as they are already present in the main text.

Author response: We have amended the supplementary materials by adding “Figure S1: Time course Cr(VI) reduction by bacterial consortia at different pH. Table S1: Total Cr and Cr(III) concentration measurements during Cr(VI) reduction using indigenous consortia bacteria”.

  1. The authors should better highlight the novelty of the present study compared to their previous work published in Water science and technology in 2008 (10.2166/wst.2008.669).

Author response: We have added lines 99-107 in the introduction. The text reads “Our previous study Molokwane et al [3] showed that sludge bacteria from the Brits plant had high Cr(VI) reduction capacity. The current study assesses how the Cr(VI) initial concentration, initial pH solution, and co-existing heavy metals affect the removal of Cr(VI) by consortia bacteria from the wastewater treatment plant. Furthermore, bacterium Cr(VI) reduction kinetics were also studied.”

  1. English language needs to be improved as there are several errors throughout the manuscript. Native speaker revision would be beneficial.

Author response: We have proofread the manuscript again, and Prof Chirwa who is an experienced scholarly writer and a native English speaker has edited this manuscript. 

Round 2

Reviewer 3 Report

The authors answered in a good way to reviewers' comments. I consequently recommend manuscript acceptance in its present form.